# Preliminary Investigation to Address Pain and Haemorrhage Following the Spaying of Female Cattle

**DOI:** 10.3390/ani10020249

**Published:** 2020-02-05

**Authors:** Audrey Yu, Dominique Van der Saag, Peter Letchford, Peter Windsor, Peter White

**Affiliations:** 1Sydney School of Veterinary Science, Faculty of Science, The University of Sydney, Camden, NSW 2570, Australia; dominique.van.der.saag@sydney.edu.au (D.V.d.S.); peter.windsor@sydney.edu.au (P.W.); p.white@sydney.edu.au (P.W.); 2Pastoral Veterinary Solutions, Kununurra, WA 6743, Australia; letchford.peter@gmail.com

**Keywords:** analgesia, animal welfare, beef cattle, behaviour, pain, spaying

## Abstract

**Simple Summary:**

The spaying of female cattle is a routine husbandry procedure conducted in some extensive beef systems, including in northern Australia. Female cattle may be spayed to control stocking rates, reduce mortalities associated with breeding, or to enable surplus females to be sold in compliance with live export requirements. The more widely practiced Willis dropped ovary technique involves severing the ovarian attachments via use of an ovariotome, which is inserted trans-vaginally to enter the abdominal cavity. While the procedure has been shown to cause pain, stress, morbidity, and mortality, it is mostly conducted without the use of veterinary pharmaceuticals. This study evaluates the efficacy of a topical anaesthetic, haemostatic wound dressing, and a non-steroidal anti-inflammatory drug for minimising pain and haemorrhage in the acute period post-spaying via the Willis dropped ovary technique. Adverse behavioural responses observed in spayed heifers were reduced in those cattle that received the non-steroidal anti-inflammatory drug, suggesting an improvement in animal welfare.

**Abstract:**

Multiple physiological and neuroendocrine changes consistent with stress and pain have been demonstrated in cattle spayed via the Willis dropped ovary technique (WDOT). The procedure is routinely conducted without the use of anaesthetics or analgesics and has major implications for animal welfare. This study aimed to evaluate the effect of a topical anaesthetic (TA), haemostatic wound dressing, and meloxicam on pain behaviour and haemorrhage in the acute period following spaying. Yearling Brahman heifers (*n* = 75) were randomly allocated to the following treatment groups: (1) rectal palpation/control (CON); (2) WDOT spay (S); (3) WDOT spay with meloxicam (SM); (4) WDOT spay with TA (STA); and (5) WDOT spay with TA and meloxicam (STAM). Individual behavioural responses, body weight, packed cell volume (PCV), and total plasma protein (TPP) were monitored for up to 24 h following treatment. Head tucking behaviour and tail stiffness was increased in all spay groups compared to the CON group (*p* < 0.001), with the lowest proportional increase in the SM group. Rumination was initially reduced in S, SM, and STA heifers compared to CON heifers (*p* < 0.001), though SM heifers ruminated more than S heifers (*p* < 0.001). CON and SM heifers stood with an arched back the least, spent the most time eating, and spent less time lying down and more time standing compared to other treatment groups (*p* < 0.001). There was no significant effect of treatment on weight change (*p* = 0.519), PCV (*p* = 0.125) or TPP (*p* = 0.799). The administration of meloxicam is suggested as an effective, currently available method for improving the welfare of cattle undergoing WDOT spaying.

## 1. Introduction

In the rangeland beef cattle production systems of northern Australia, female cattle that are not required for breeding are spayed to control stocking rates and to enable surplus females, particularly heifers and aged animals, to be sold [1,2,3]. In addition, spaying reduces breeder mortalities, which is of particular economic importance as intact heifers may die from complications associated with pregnancy, parturition, and lactation [3]. Surgical spaying is currently the only practical and reliable method for rendering females sterile in extensively managed beef herds, and is also performed in southern Africa, and North America, and South America [2]. In these regions, bulls and females may be continuously grazed together as the labour and cost involved with fencing is prohibitive given property size, with breaches caused by seasonal bushfires or floods [2,3,4]. Females must also be non-pregnant in compliance with the strict requirements of Australia’s live export trade with south-east Asia [5].

The three methods used for the routine spaying of cattle are flank laparotomy (FL), the Willis dropped ovary technique (WDOT), and passage spaying/webbing [1,2,6]. In the former, cattle are electrically immobilized, a flank incision made through the abdominal wall, and either the ovaries or a portion of the oviduct are excised [5]. Spaying via the WDOT involves the use of an ovariotome, a stainless steel rod with a flattened spear head end containing a cutting slot [7,8]. The tool is inserted into the vagina and pierces the anterior vaginal wall dorsal to the cervix [8]. The ovaries are in turn manipulated by rectal palpation into the cutting slot of the ovariotome and their attachments are severed by retracting the instrument, which drops the ovaries into the abdominal cavity [8]. Passage spaying or webbing involves the removal of the ovary or oviduct via a small incision through the vaginal wall allowing two fingers to access these structures [6]. This procedure is limited to either primigravida or larger well grown heifers [6]. Although limited, much of the existing research evaluating the physiological, behavioural, and neuroendocrine responses of cattle to spaying has demonstrated that both FL and the WDOT cause pain and stress [1,2,5]. However, the WDOT is preferable as it causes less pain and stress, it reduces the risk of infection, it does not result in hide or carcass damage, and animals recover faster, which allows them to be marketed sooner [1,2,5,8]. In addition, faster processing rates are achievable using the WDOT, with skilled operators spaying up to 600 cattle per day [5,8]. However, potential complications from WDOT spaying include internal haemorrhage from the ovariectomy site, intestinal or rectal perforation, and peritonitis [5,8], all of which can lead to mortality rates of approximately 0.5%–1.5% [5]. Currently, WDOT spaying is conducted routinely without the use of anaesthetics or analgesics [1,2,5].

Pain management is a progressively important consideration in livestock production given its role in improving animal welfare outcomes [9]. Multimodal analgesia is the use of analgesic drugs with different mechanisms of action, and incorporating a local anaesthetic with a non-steroidal anti-inflammatory drug (NSAID) has been shown to be the most effective way of mitigating both acute and prolonged phases of pain after common husbandry procedures in cattle, such as castration and dehorning [10,11,12,13,14]. In Australia, a multifunctional topical anaesthetic, antiseptic, and haemostatic wound dressing (Tri-Solfen^®^; Bayer Animal Health, Australia) is registered for use in lambs and calves undergoing routine husbandry procedures. It contains the anaesthetic agents lignocaine (40.6 g/L) and bupivacaine (4.5 g/L), as well as cetrimide (5.0 g/L) and adrenaline (24.8 mg/L). The wound dressing is applied post-operatively to exposed wounds and effects rapid and prolonged pain alleviation, as observed through a reduction in pain-related behaviours [15,16]. This gel-based topical anaesthetic is practical for use on-farm as it is quick and easy to apply, addressing many of the constraints associated with injectable forms of local anaesthetic. For clarity, Tri-Solfen^®^ and all its components will now be referred to as topical anaesthetic (TA). The NSAID meloxicam is another effective analgesic option available for use in cattle, which may be administered as an injection or orally following routine painful procedures for a prolonged duration of effect [12]. The use of practical means of pain alleviation for spaying could have major implications for improving cattle welfare.

The current study aimed to evaluate the effect of a topical anaesthetic, haemostatic wound dressing, and meloxicam, alone and in combination, on pain and haemorrhage in the acute period following spaying of heifers via the WDOT. The primary objectives were to assess pain via behavioural observation, and to evaluate post-operative haemorrhage through an assessment of packed cell volume (PCV) and total plasma protein (TPP). It was hypothesised that both the TA and meloxicam would reduce pain following spaying, with a combination of the two being most effective, and that the adrenaline included in the TA would reduce haemorrhage following spaying.

## 2. Materials and Methods

### 2.1. Animal Ethics

The experimental protocol was approved by the University of Sydney Animal Ethics Committee (approval no. 1258). The study was conducted in compliance with national standards for the care and welfare of animals.

### 2.2. Location and Environmental Conditions

The experiment was conducted on a commercial beef cattle property located approximately 39 km north-west of Kununurra in the East Kimberley region of Western Australia (15°29′11″ S, 128°32′1″ E). The experiment occurred over a 3-day period in the early wet season (late November 2017). The daily minimum and maximum temperature ranges during the experimental period were between 23.5 and 26.2 °C and 36.8 and 39.8 °C, respectively. The average relative humidity ranged from 39% to 69%, and there was a total of 7.4 mm rainfall.

### 2.3. Animals, Treatments, and Experimental Design and Conduct

The yearling Brahman heifers (*n* = 75, body weight 260.8 ± 16.5 kg) selected for spaying were representative of cattle routinely spayed in the region. Animals were mustered to the yards two days prior to the experimental period. One of the heifers included had been recently dehorned (within the two weeks preceding the study) and none were found to be pregnant using rectal palpation during the experiment.

A Willis ovariotome (Willis Spay Tool, Bainbridge Pty Ltd., Murarrie, Qld, Australia) was modified by a local engineering company prior to commencement of the experiment. The modified ovariotome featured a thin, cylindrical metal tube attached laterally parallel to the length of the tool, and opening at the distal end near the cutting slot (Figure 1). On the operator side, a rubber tube was attached to the modified segment of the ovariotome and acted as an extension line through which topical anaesthetic could be administered by syringe. All manipulative and surgical procedures were performed by a highly skilled, experienced veterinarian who routinely performs the WDOT spay procedure on 20,000 cattle per annum.

The experiment was conducted in a set of steel commercial cattle yards and adjacent holding yards. Three holding yards were used, each approximately 20 by 25 m (Yards 1, 2 and 3). The procedures started at 0630 h, with three replicates of 14 heifers (2–3 of each treatment), and 2 replicates of 14 and 19 heifers (2–4 of each treatment) processed respectively on two successive days (Table 1). This protocol was designed to ensure that, on the day that procedures were conducted (day 1 and day 2), there would be sufficient hours of daylight after the last treated animal for a minimum 6 h of behavioural observation.

On day 1, the cattle within each replicate group were moved from the holding yard into a crush via a set of scales where animals were weighed immediately prior to being restrained for the procedure. A blood sample was taken from the tail vein, and the treatment, as previously assigned to each individual animal, was performed. After the procedure, heifers were head-bailed, ear-tagged (left ear), and a corresponding number was spray-painted on the flank bilaterally for individual identification. The animal was then released into an adjacent holding yard.

Once all cattle within the replicate had been sampled and treated, the group was moved to another yard to allow for the next replicate group. This was repeated until the 3 replicates were completed on day 1. In each yard, feed and water were provided ad libitum with good quality pasture hay and 1–2 water troughs per yard. The following morning (day 2), the cattle from day 1 were observed for an hour at the start of the day, approximately 24 h post-treatment. All replicate groups were then moved through the crush and individually restrained for repeat weighing and blood sampling. Cattle were then released into a larger holding yard where they were kept for a week following spaying for observation by staff to detect any post-operative mortalities. All animals were then moved to a paddock to graze as a single group with ad libitum pasture, hay, and water. These procedures were repeated for the 2 replicate groups on day 2.

Each animal was randomly allocated to 1 of 5 treatment groups, with 14-16 heifers assigned to each of the treatments. The treatments were: (1) rectal palpation (CON, *n* = 14); (2) a WDOT spay (S, *n* = 15); (3) a WDOT spay with intra-operative delivery of meloxicam (SM, *n* = 16); (4) a WDOT spay with intra-operative delivery of TA (STA, *n* = 15); and (5) a WDOT spay with intra-operative delivery of TA and meloxicam (STAM, *n* = 15). The WDOT involves rectal palpation and manipulation of the reproductive tract prior to the insertion of a device into the vagina. Hence, we examined the responses of cattle to this initial, less invasive component of cattle spaying procedures in the control group. The procedures performed are detailed below.


*(1) Rectal palpation (CON)*


Each heifer was physically restrained in a commercial cattle crush, with the kick-gate closed behind the back legs. Rectal palpation was performed with brief manipulation of the reproductive tract (10–15 s) [17]. The entire procedure was completed within 1 min.


*(2) WDOT spay using modified ovariotome (S)*


The heifers were physically restrained as described above. Cattle were spayed according to procedures described by de Witte, Jubb, and Letchford [18], using the modified ovariotome. The tool was disinfected between uses by immersion in iodine, prior to being inserted into the vagina and puncturing through the vaginal fornix to enter the caudal abdominal cavity. Each ovary was manipulated by transrectal palpation into the cutting slot of the ovariotome and the attachments severed. After the procedure, heifers were head-bailed and a 15 mm diameter hole was punched in the pinna of the left ear using standard spay ear-punch pliers in accordance with livestock legislation. On average, the entire procedure was performed within 3 min.


*(3) WDOT spay using modified ovariotome with intra-operative delivery of meloxicam (SM)*


The heifers were physically restrained and the spay was performed as described above. An injection of meloxicam (Metacam^®^ 20 mg/mL; Boehringer Ingelheim Vetmedica, Germany) at a dose of 0.5 mg/kg body weight was administered subcutaneously immediately after the procedure. Doses were pre-calculated for body weight groups at 50 kg intervals from 150–400 kg to facilitate ease of drug administration and to minimise the overall handling time of animals. On average, the entire procedure was performed within 3 min.


*(4) WDOT spay with delivery of TA *via* modified ovariotome (STA)*


The heifers were physically restrained and the spay was performed as above, with the addition of Tri-solfen^®^ TA administered at two key points during the procedure by an assistant: prior to piercing through the vaginal wall, and immediately prior to ovary excision. These were identified to be the most invasive and likely more painful parts of the procedure. Approximately 6–8 mL of TA were administered per heifer. On average, the entire procedure was performed within 5 min.


*(5) WDOT spay with the delivery of TA *via* modified ovariotome and the intra-operative delivery of meloxicam (STAM)*


As described above for the SM and STA procedures, cattle received a combination of TA during the procedure and meloxicam intra-operatively.

### 2.4. Blood Sampling and Assaying

A blood sample was taken from each heifer immediately following restraint in the crush, prior to treatment. Approximately 10 mL of blood was collected from the tail vein into labelled lithium heparin or K_2_ EDTA 18 mg Plus Vacutainer tubes (Becton-Dickinson, Plymouth, UK). A repeat sample was obtained the following morning at approximately 24 h post-treatment. The blood samples were placed in a portable refrigerator maintaining an ambient temperature below 10 °C until all replicates for the day were completed. The samples were then centrifuged using a 24 place microhaematocrit centrifuge at RCF 13,000× *g* for 7 min at room temperature (Haematospin 1400, Hawksley, Sussex, UK). The PCV and TPP for each animal sampled that morning was recorded [19].

### 2.5. Behavioural Recordings

Video recordings of heifers in the holding yards were taken in order to observe and comprehensively quantify the behavioural responses following spaying with minimal disturbance by human activity. On day 1, four HD 1080p Sports Action Cam video cameras (Sony Australia Ltd., North Sydney, NSW, Australia) were mounted on Yards 1 and 2 in each corner, and 3 cameras were mounted on Yard 3 (Figure 2). As there were only two yards in use on days 2 and 3, the additional cameras from Yard 3 were mounted in Yard 1 and 2. Two additional cameras were mounted in Yard 1 and one was added to Yard 2 (Figure 2). Recording began when the cattle were released into the yards immediately following the procedure, and continued for the following 6 h (Observations 1–6). Cattle were also observed for an hour at the start of the following day at approximately 24 h post-treatment (Observation 7), prior to processing the day 2 replicates. Morning observations on day 2 and day 3 commenced as soon as it was light enough to discern flank numbers (approximately 0530). The videos were analysed retrospectively following the completion of the experiment using The Observer^®^ XT 13 observational data software program (Noldus Information Technology, Wageningen, The Netherlands).

Seven observations were recorded in total for each replicate group, corresponding to time after procedures had been conducted. Observation 1 was recorded one hour after the procedure was performed, and, similarly, Observations 2–6 were documented their respective hours after the procedure (between 0730 and 1700). Observations were analysed by a single observer to standardize the recordings, and the observer was blinded to the treatments. For each observation (O1-7), each animal was observed over a 10 min period within the hour. At one minute intervals, the behaviours exhibited by the animal were recorded, giving a total of 10 instantaneous samples reflective of the frequency of each behaviour. The key behaviours analysed (Table 2) were derived from an ethogram developed by a cattle behaviour expert (J. C. Petherick), with definitions focusing on movements or motor patterns to avoid inherent bias associated with interpretation [20]. Visibility of cattle in the yards was impeded at times, hence the number and frequency of observations for each replicate group varied. Recordings occurred in an order depending on the observer’s ability to view the animal’s flank number.

### 2.6. Morbidity and Mortality Recordings

The general health status of the heifers was visually assessed and monitored daily for a week following the procedures by property staff. All mortalities were systematically examined by necropsy by the attending veterinarian.

### 2.7. Statistical Analyses

For the behavioural analyses, data was exported from The Observer XT13 programme and all analyses were conducted using Genstat statistical software (18th Edition VSN International Ltd., Hemel Hempstead, UK). Data from the first 6 h on frequency of behaviours were combined to provide an improved overall assessment of post-procedure behaviour. Data from the 24 h timepoint was assessed separately. Regression analysis was performed with generalised linear modelling, using binomial proportions. A macro in Microsoft Excel 2013 was then used to generate the odds and probabilities for each behaviour. Heifer 47 died on day 3 of the experiment and was hence removed from all statistical analysis to avoid any bias. As not all cattle were located at every sample, for data analysis, results were expressed as proportions of the number of observations.

As animals were not divided into treatment groups according to weight, an ANOVA was also performed to test for significant differences in the effect of treatments in relation to initial body weights. Data on changes in weight, PCV, and TPP at 0 h and 24 h for each animal was entered into Excel 2013. Data was subjected to linear mixed modelling (REML) to account for individual animal variation and to determine if time or treatment had a significant effect on these factors. Outliers were identified using boxplots. Outliers were removed from the data regarding changes in weight and the data was re-analysed, as this was likely due to inaccurate scale readings rather than a true dramatic change in weight. For all statistical calculations, *p* values ≤ 0.05 were considered statistically significant.

## 3. Results

### 3.1. Morbidity and Mortality

One mortality was documented during the experiment, where a STAM heifer treated on day 1 was found dead on arrival on the morning of day 3. A post-mortem was performed, and findings included free blood in the abdomen, a larger left ovarian pedicle stump and corresponding blood clot indicating increased haemorrhage from the area previously. The puncture point in the vaginal wall was skewed to the left into the broad ligament—a structure with increased vascularity. The mortality occurred as a result of internal haemorrhage, though it was unclear whether haemorrhage from the ovarian pedicle stump or the broad ligament was the primary cause.

### 3.2. Changes in Weight, PCV, and TPP

No significant differences were found between treatments in terms of initial body weight (*p* = 0.392). There was no significant effect of treatment on PCV (*p* = 0.125), TPP (*p* = 0.799), or weight (*p* = 0.519) from 0 to 24 h. Outliers were identified in each data set. Heifers 87 (STAM), 88 (SM), and 67 (SM) had marked increases in post-procedure PCV, while heifer 65 (STA) had a marked decrease in PCV. Heifer 86 (STA) had a marked increase in TPP and a marked decrease in weight. Heifer 56 (STAM) had a marked increase in weight.

### 3.3. Behavioural Analysis

The numbers of heifers able to be identified for each hourly observation varied from 0 to 18 individuals.

Between 1 and 6 h, there was a significant effect of treatment on lying, standing, repetitive head movement, head tuck, head down, head up, back arch, repetitive tail movement, relaxed tail, and stiff tail behaviours (*p* < 0.001), as well as on walking (*p* = 0.047) and self lick behaviours (*p* = 0.002). There were insufficient recordings of drinking and licking other for statistical analysis.

At the 24 h timepoint, there was a significant effect of treatment on lying, standing, head down, repetitive tail movement, relaxed tail, and stiff tail behaviours (*p* < 0.001), as well as on walking (*p* = 0.036). There was no significant effect at this timepoint of treatment on repetitive head movement, head tuck, self lick, or back arching behaviours.

#### 3.3.1. Lying, Standing, and Walking

The results for this are summarised in Table 3. Between 1 and 6 h, S, STA, and STAM heifers spent significantly more time lying down compared to CON and SM heifers (*p* < 0.001). The SM and CON heifers spent the least amount of time lying down. The STA heifers spent the greatest proportion of time lying down. The reciprocal for this pattern was shown for standing, with S, STA, and STAM heifers spending significantly less time standing compared to CON and SM heifers (*p* < 0.001). CON and SM heifers stood for the longest duration, and STA heifers spent the least proportion of time standing. SM and STAM heifers spent a significantly greater proportion of time standing compared to S heifers (*p* < 0.001). STA heifers spent significantly less time walking than S and STAM heifers (*p* = 0.047).

At 24 h post-procedure, the SM and STAM groups spent a significantly greater amount of time lying down compared to the CON and S groups (*p* < 0.001). The S and CON heifers spent the least amount of time lying down, and STAM heifers spent the most time lying down. SM, STA, and STAM heifers spent significantly less time standing than CON and S heifers (*p* < 0.001). Significantly more STA heifers were observed walking compared to CON and STAM groups (*p* = 0.036).

#### 3.3.2. Head Position and Movement

The results for this are summarised in Table 4. Between 1 and 6 h, CON heifers stood with their head down significantly more than S, STA, and STAM heifers (*p* < 0.001). SM heifers stood with their head down significantly more than S heifers, while STA heifers stood with their head down for significantly less time compared to S heifers (*p* < 0.001). Conversely, STA heifers stood with their head up for a greater proportion of time compared to both the CON and S groups (*p* < 0.001). S and SM heifers exhibited significantly more repetitive head movements compared to CON heifers, while STA heifers were observed performing this behaviour significantly less frequently (*p* < 0.001). The frequency of head tucking behaviour was significantly increased in all groups compared to the CON group (*p* < 0.001), with the highest frequency in the STA and STAM groups. SM, STA, and STAM heifers exhibited self-licking behaviour significantly less frequently than S heifers (*p* = 0.002).

At the 24 h timepoint, SM, STA, and STAM heifers spent significantly less time standing with their head down compared to CON heifers (*p* < 0.001). SM heifers stood with their head down the least. Reciprocally, SM, STA, and STAM heifers stood with their head up for a greater proportion of time compared to CON heifers (*p* < 0.001).

#### 3.3.3. Back and Tail Position and Movement

The results for this are summarised in Table 5. Between 1 and 6 h, S, STA, and STAM heifers stood with an arched back significantly more than CON and SM heifers (*p* < 0.001). CON and SM heifers stood least frequently with an arched back, and STA heifers stood with an arched back most frequently. In all treatment groups (S, SM, STA, STAM), heifers stood with a stiff tail for significantly longer than heifers in the CON group (*p* < 0.001). STA heifers stood the greatest proportion of time with a stiff tail, while SM heifers stood for the least amount of time with a stiff tail out of the spayed groups. Conversely, CON heifers spent the greatest proportion of time with a relaxed tail. There was a significant reduction in the proportion of heifers with a relaxed tail in all treatment groups compared to the CON group (*p* < 0.001). SM and STAM heifers had significantly increased repetitive tail movements post-procedure compared to CON heifers (*p* < 0.001). STA heifers spent the least proportion of time with repetitive tail movements, followed by CON and S heifers.

At 24 h post-procedure, S heifers spent a significantly greater proportion of time with a stiff tail compared to all other treatment groups (*p* < 0.001). CON heifers stood with a stiff tail for the least amount of time, and spent the greatest proportion of time with a relaxed tail. There was a significant reduction in the proportion of heifers with a relaxed tail in all treatment groups compared to the CON group (*p* < 0.001). Significantly more repetitive tail movements were observed in SM, STA, and STAM groups compared to the CON group (*p* < 0.001). SM heifers showed the most repetitive tail movements proportionally, while CON heifers showed the least.

#### 3.3.4. Eating and Ruminating

The results for this are summarised in Table 6. Between 1 and 6 h, S, STA, and STAM heifers spent significantly less time eating compared to CON and SM heifers (*p* < 0.001). The CON heifers spent the most amount of time eating, followed by SM heifers. The STA heifers spent the least amount of time eating. In addition, CON heifers spent the greatest proportion of time ruminating. Rumination was significantly reduced in S, SM, and STA heifers compared to CON heifers (*p* < 0.001). Compared to S heifers, SM and STAM heifers ruminated significantly more, while STA heifers ruminated less (*p* < 0.001).

At the 24 h timepoint, all treatment groups spent a significantly reduced amount of time eating compared to the CON group (*p* < 0.001). Significantly less rumination was observed in S heifers (*p* < 0.001), while rumination was significantly increased in SM and STAM heifers compared to CON heifers (*p* < 0.001).

## 4. Discussion

The findings from this study indicate that the WDOT spaying of cattle does negatively impact animal welfare with behavioural responses indicative of discomfort and pain for at least 6 h following the procedure. Various behavioural changes were observed in the time spent lying, standing, walking, eating and ruminating, as well as displays of repetitive head movement, head tucking, self licking, back arching, tail stiffness, and repetitive tail movement. The cattle that received meloxicam immediately after spaying exhibited a reduced incidence of behaviours indicative of pain, whereas the cattle that received TA during spaying exhibited behaviours indicative of pain. The results on the ability of TA to minimise haemorrhage locally were not conclusive. This is the first study to examine the effects of analgesia on the welfare outcomes of cattle undergoing spaying. The findings suggest that pain can be relieved during the acute post-operative period through the use of meloxicam.

WDOT spaying is associated with mortality rates of 0.5%–1.5% [5], with one mortality documented in this experiment. While ovariotomes are available in a range of sizes to suit different ages [8], a modified cow ovariotome was used in the experiment and this may have caused greater trauma in heifers as the tool is larger and blunter, with a more rounded head. The vet performing the spay procedures noted that this also made it more difficult to maintain the ovariotome in place at the ideal puncture site through the vaginal wall, where it is least vascular (P Letchford, pers. comm.). The mortality occurred within 24–48 h of the procedure being performed, as with most deaths related to surgical haemorrhage, though mortalities can occur up to 7 days after spaying from acute diffuse peritonitis [5]. The post-mortem findings were consistent with abdominal haemorrhage, likely after a clot rupture at the ovarian pedicle stump.

The findings on post-spaying behaviour from the current experiment are consistent with documented responses to pain in ruminants [2,3,5,7,13,21,22,23,24]. In the 6 h post-procedure, spayed heifers spent more time lying down and less time standing, eating, and ruminating, exhibited increased head tucking behaviour, as well as repetitive head and tail movements, and stood more frequently with an arched back and a stiff tail compared to CON heifers (Figure 3). An initial study in North America on *Bos Taurus* heifers reported mild stiffness and straining in spayed animals in the first 12 h following surgery, with stiff walking observed on the following day also [7]. Head turning is believed to be similar to flank-watching in horses with colic [22,23], while tail flicking can be a response to irritation, local painful stimuli, or flies [13,21,24] and has been associated with pain from castration [13,21,22,23,24]. The suppression of appetite and rumination can occur subsequent to pain in spayed cattle [2,5]. An increased incidence of tail stiffness and a reduction in eating and rumination were also observed at 24 h post-procedure in spayed heifers, indicating prolonged pain. In this study, it appears that the reluctance to move may also be indicative of pain, as observed with the S, STA, and STAM groups, which spent the most time lying down [25], and spaying has previously been associated with increased time standing, with the head down, or lying down [2]. In contrast, excessive locomotion, such as increased walking, has also been an observed pain response in dehorned and castrated calves [26]. The effects of pain on locomotion, standing, and lying behaviour can vary, and it appears that different noxious stimuli may elicit unique behavioural responses. A study by Petherick et al. (2013) [2] observed that more WDOT heifers stood with their head down compared to non-spayed heifers for up to 3 days after treatment. In contrast, the CON group in the current study stood with their head down the most, with significantly fewer S, STA, and STAM heifers exhibiting this behaviour. This behaviour is difficult to interpret as many animals had their head down while eating, and thus eating may have been a confounding factor in the analysis of head down behaviour. CON and SM groups spent the greatest proportion of time eating and also stood with their head down the most, while S, STA, and STAM groups had reduced feeding post-procedure, suggesting increased pain in these heifers.

These adverse behavioural responses were less pronounced in the meloxicam treatment group (SM), which often had similar results to the CON group, indicating that meloxicam is effective at mitigating the pain and discomfort associated with spaying for at least 6 h post-procedure. Non-steroidal anti-inflammatory drugs function by inhibiting cyclooxygenase (COX) enzymes, resulting in a decreased production of prostaglandins and a reduced inflammatory response [27]. Meloxicam is a preferential COX-2 inhibitor, and its use has been associated with a reduction in plasma cortisol concentration and substance P, a pain neurotransmitter, in castrated and dehorned calves [24,28,29]. In the first 6 h, CON and SM heifers consistently spent the least amount of time lying down and the most time standing, stood least frequently with an arched back, and spent the most time eating (Figure 3). Compared to S heifers, SM heifers ruminated significantly more between 1 and 6 h and at 24 h post-procedure (Figure 3). While the frequencies of head tucking and tail stiffness were significantly increased in all treatment groups compared to the CON group between 1 and 6 h, the lowest proportional increase out of the treatment groups was shown in the SM group (Figure 3). These results suggest that meloxicam can be effectively used in cattle undergoing WDOT spaying. However, proportionately, SM heifers still spent more than double the amount of time with a stiff tail compared to CON heifers and had increased repetitive head and tail movements. This may suggest post-operative treatment with meloxicam alone does not completely alleviate the discomfort or pain caused by spaying, and supports previous data on the pain of castration and NSAID use in cattle whereby peri-operative nociception is not completely abolished [10,14,30].

The use of TA, both alone and in combination with meloxicam, increased the incidence of pain-related behaviours after spaying. TA has been shown to achieve rapid wound anaesthesia within 1 min and is effective in the amelioration of pain for up to 24 h after surgical castration in calves [16]. In this study, the use of TA appeared to cause adverse effects, with results contradicting the original hypothesis that coating the vaginal wall and ovaries with the TA gel prior to excision would effectively mitigate acute pain. STA heifers displayed behaviours consistent with significant discomfort, spending the greatest proportion of time lying down and the least proportion of time standing, walking, eating, and ruminating (Figure 3). The highest frequency of back arching and tail stiffness also occurred in this group, while head tucking occurred most frequently in STA and STAM groups (Figure 3). Given that Tri-solfen^®^ is licensed for use on external wounds, it is possible that cetrimide, an antiseptic contained in the product, is irritant to the abdominal viscera with resultant chemical peritonitis. The TA gel was applied intra-vaginally and intra-abdominally prior to ovariectomy, and may have caused local inflammatory reactions resulting in increased pain behaviours. Interestingly, STA heifers demonstrated the least repetitive head and tail movements from all groups, and the lack of these behaviours could potentially be reflective of a strong pain response. Behavioural responses of the STAM group were often intermediate between STA and CON/S groups, which may be attributed to the analgesic effects of meloxicam countering any local irritation from Tri-solfen^®^.

While multimodal analgesia is ideal for optimising pain management, the use of the meloxicam and Tri-Solfen^®^ combination in this experiment did not yield better results than when meloxicam was administered solely. The effectiveness of a local anaesthetic and NSAID combination on pain mitigation has been well documented [14,24], and has been shown to be a more effective protocol compared to use of NSAIDs or local anaesthetic alone [14]. For example, the administration of lignocaine with either of the NSAIDs flunixin meglumine or ketoprofen has been shown to prevent an increase in plasma cortisol concentration and, by inference, pain and distress after castration in calves [11,13,14]. Notably, in the current study, all spayed heifers spent a significantly reduced amount of time eating compared to the CON group at the 24 h timepoint (Figure 3). Appropriately managing pain in cattle may reduce the effects of stress responses, immune reactions, and reduced feeding behaviour, all of which affect production parameters, including weight gain [31]. For example, calves undergoing castration and/or dehorning with a local anaesthetic and NSAID have been shown to spend more time eating compared to calves that did not receive any analgesics [13], with improved short-term weight gain [30]. Further research into an effective multimodal pain relief protocol is suggested.

It was also hypothesised that the adrenaline contained in Tri-solfen^®^ could assist in local vasoconstriction and potentially minimise haemorrhage from severing the ovarian pedicle. However, no significant difference was found between the PCV and TPP readings of CON and spayed animals at 0 and 24 h; therefore, spaying via the WDOT may only cause severe haemorrhage in few cases.

In order for the development and approval of effective analgesic drug protocols for use in livestock, validated, repeatable methods of measuring pain must first be established [10]. However, pain assessment in cattle can be particularly challenging as prey species inherently conceal pain to limit vulnerability [32]. The objective monitoring of behavioural changes through the use of devices such as accelerometers could be a consideration to provide unbiased, validated data in future research [24]. A larger sample size would also allow analysis of hourly behavioural trends to determine the duration of the efficacy of various analgesics and anaesthetics that are commonly used. While the current study was limited to heifers, pain and stress have been shown to be more severe and sustained in spayed cows [2], and this should be considered for further research. Until practical, less invasive alternatives that can effectively control reproduction in extensive rangeland systems are developed, there should be a focus on addressing immediate welfare considerations with the use of appropriate veterinary pharmaceuticals.

## 5. Conclusions

This study demonstrates the animal welfare implications of spaying cattle without the use of pain relief, as it occurs according to current industry practice and state regulations. There were significant behavioural changes in WDOT-spayed heifers consistent with discomfort and pain in the initial 6 h and up until 24 h post-procedure. It appears that meloxicam is effective at mitigating the pain and discomfort associated with spaying, and could be practically incorporated into routine operations for the improvement of animal welfare. Further research into an effective multimodal analgesic protocol and haemorrhage prevention for spaying are warranted, with alternate methods for administering a local anaesthetic agent solely being suggested for future investigations.

## Figures and Tables

**Figure 1 animals-10-00249-f001:**
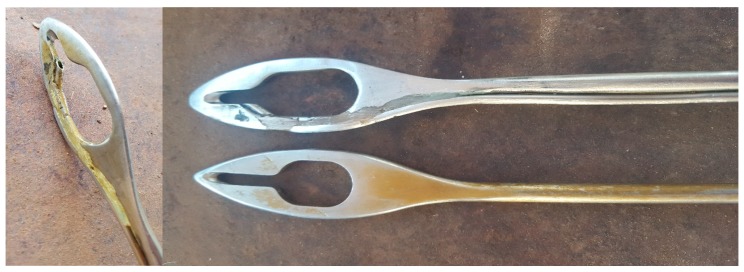
Modified Willis ovariotome (left and top right), with cylindrical metal tube attached parallel to the length of the tool, opening near the cutting slot. Willis ovariotome, unmodified (bottom right).

**Figure 2 animals-10-00249-f002:**
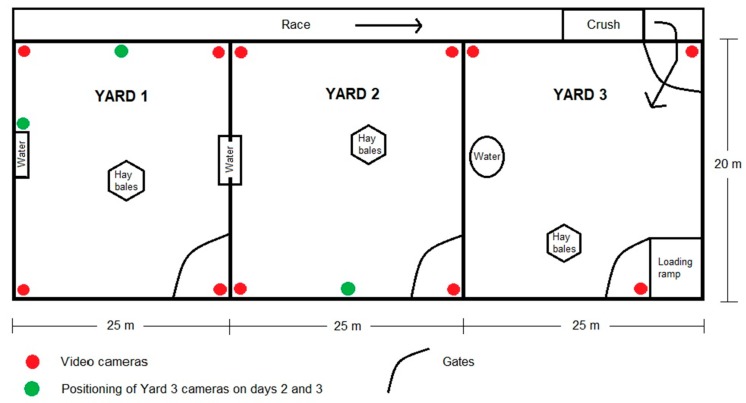
Model illustration of yard layout and video camera positioning.

**Figure 3 animals-10-00249-f003:**
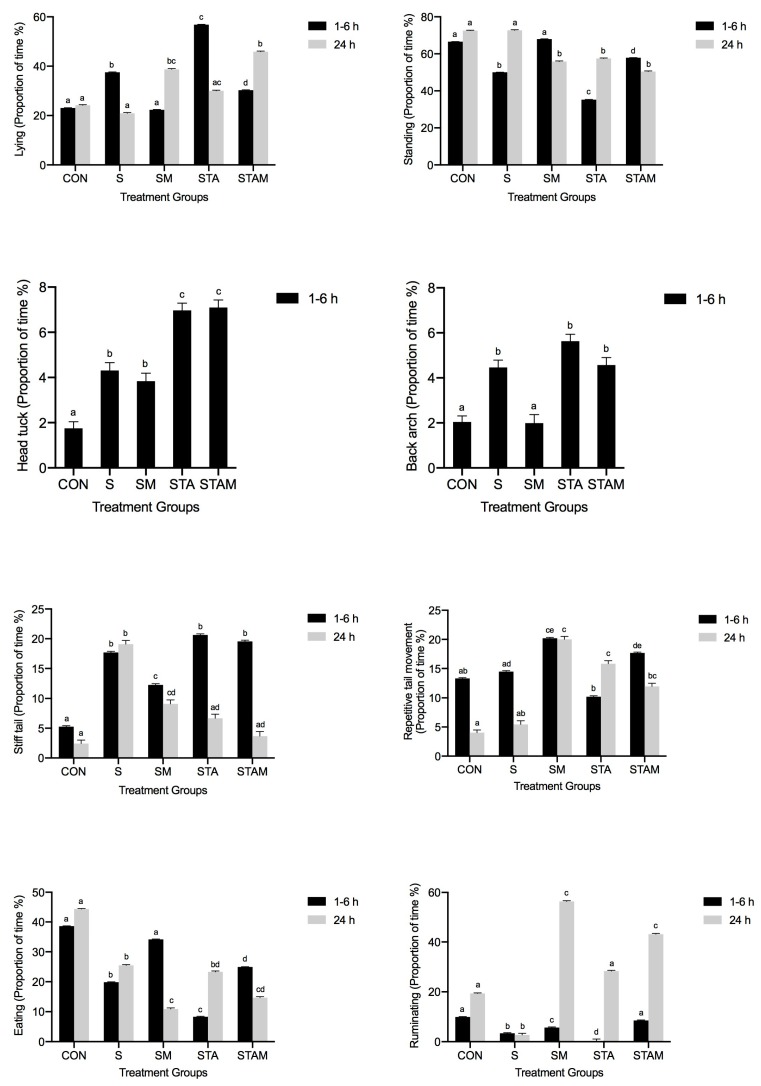
Comparison of select behaviours exhibited by heifers at 1–6 h and 24 h post-procedure after rectal palpation (CON) or after spaying with the Willis dropped ovary technique without anaesthesia or analgesia (S), with intra-operative delivery of meloxicam (SM), a topical anaesthetic (STA), or a combination of both treatments (STAM). Columns with different letters for the 1–6 h and 24 h timepoints are significantly different (*p* ≤ 0.05).

**Table 1 animals-10-00249-t001:** Number of heifers in each treatment group by yard.

Day	Yard	Treatment					Total Heifers
		CON	S	SM	STA	STAM	
1	1	2	3	3	3	3	14
	2	3	2	3	3	3	14
	3	3	3	3	2	3	14
2	1	3	3	3	3	2	14
	2	3	4	4	4	4	19

CON = rectal palpation; S = Willis dropped ovary technique spay; SM = Willis dropped ovary technique spay with intra-operative delivery of meloxicam; STA = Willis dropped ovary technique spay with intra-operative delivery of topical anaesthetic; and STAM = Willis dropped ovary technique spay with intra-operative delivery of topical anaesthetic and meloxicam.

**Table 2 animals-10-00249-t002:** Ethogram developed for behavioural observations of heifers following rectal palpation and Willis dropped ovary technique spaying with or without meloxicam and topical anaesthetic, modified from Petherick et al. [2].

Behaviour		Description
Standing, walking	Head down	Head level with or below brisket
	Head up	Head above brisket
	Head tuck	Head turned laterally towards flank
	Rapid/repetitive movement of head	Rapid tuck movement or repeated head shaking
	Tail relaxed	Tail resting against body, not elevated
	Tail stiff	Tail held stiffly away from body
	Repetitive movement of tail	Repeated tail flicking
	Back flat	Back level with or below withers
	Back arch	Back elevated above the level of the withers
Lying		Lying on sternum or partially on sternum with hindquarters to one side; lying on side, fully recumbent
Feeding		Taking hay into mouth and/or chewing hay and/or grazing/browsing
Drinking		Consuming water
Ruminating		Regular chewing and regurgitation movements
Licking	Self lick	Licks flank or other body part
	Lick other	Licks another animal

**Table 3 animals-10-00249-t003:** Proportion of time spent lying down, standing and walking (±SE) by heifers in each treatment group at 1–6 h and 24 h post-procedure.

Behaviour	Time (h)	Proportion of Time (%)					*p*-Value
		CON	S	SM	STA	STAM	
Lying	1–6	23.07 ^A^ ± 0.09	37.54 ^B^ ± 0.12	22.33 ^A^ ± 0.13	56.84 ^C^ ± 0.12	30.28 ^D^ ± 0.13	<0.001
	24	24.19 ^A^ ± 0.21	20.91 ^A^ ± 0.31	38.73 ^BC^ ± 0.29	30.00 ^AC^ ± 0.29	45.86 ^B^ ± 0.28	<0.001
Standing	1–6	66.57 ^A^ ± 0.08	50.00 ^B^ ± 0.11	67.99 ^A^ ± 0.11	35.26 ^C^ ± 0.11	57.88 ^D^ ± 0.11	<0.001
	24	72.58 ^A^ ± 0.20	72.72 ^A^ ± 0.29	55.86 ^B^ ± 0.28	57.50 ^B^ ± 0.27	50.46 ^B^ ± 0.28	<0.001
Walking	1–6	10.36 ^AC^ ± 0.13	12.46 ^A^ ± 0.17	9.67 ^AC^ ± 0.18	7.91 ^BC^ ± 0.19	11.82 ^A^ ± 0.18	0.047
	24	3.23 ^AC^ ± 0.51	6.36 ^BC^ ± 0.64	5.40 ^BC^ ± 0.66	12.50 ^B^ ± 0.58	3.67 ^AC^ ± 0.72	0.036

CON = rectal palpation; S = Willis dropped ovary technique spay; SM = Willis dropped ovary technique spay with intra-operative delivery of meloxicam; STA = Willis dropped ovary technique spay with intra-operative delivery of topical anaesthetic; and STAM = Willis dropped ovary technique spay with intra-operative delivery of topical anaesthetic and meloxicam. *p*-value for the main effect of treatment. ^A,B,C,D^ Values within a row that do not share a superscript differ significantly at *p* ≤ 0.05.

**Table 4 animals-10-00249-t004:** Proportion of time spent exhibiting various head positions or movements (±SE) by heifers in each treatment group at 1–6 h and 24 h post-procedure.

Behaviour	Time (h)	Proportion of Time (%)					*p*-Value
		CON	S	SM	STA	STAM	
Head up	1–6	69.05 ^A^ ± 0.08	72.45 ^A^ ± 0.12	67.99 ^A^ ± 0.12	81.23 ^B^ ± 0.13	70.82 ^A^ ± 0.12	<0.001
	24	56.45 ^A^ ± 0.18	64.53 ^AB^ ± 0.27	82.73 ^C^ ± 0.31	74.99 ^BC^ ± 0.28	75.23 ^BC^ ± 0.29	<0.001
Head down	1–6	25.11 ^A^ ± 0.09	15.54 ^B^ ± 0.14	20.91 ^AD^ ± 0.13	9.65 ^C^ ± 0.15	17.98 ^BD^ ± 0.14	<0.001
	24	40.32 ^A^ ± 0.18	31.82 ^AB^ ± 0.28	8.18 ^C^ ± 0.39	22.50 ^BD^ ± 0.29	16.51 ^CD^ ± 0.32	<0.001
Repetitive head movement	1–6	4.09 ^A^ ± 0.19	7.69 ^B^ ± 0.24	7.25 ^B^ ± 0.24	2.14 ^C^ ± 0.32	4.10 ^A^ ± 0.28	<0.001
	24						0.418
Head tuck	1–6	1.75 ^A^ ± 0.29	4.31 ^B^ ± 0.35	3.84 ^B^ ± 0.35	6.97 ^C^ ± 0.32	7.10 ^C^ ± 0.33	<0.001
	24						0.511
Self lick	1–6	1.02 ^AC^ ± 0.38	2.31 ^AB^ ± 0.46	0.43 ^C^ ± 0.69	0.13 ^C^ ± 1.07	0.47 ^C^ ± 0.69	0.002
	24						0.947

CON = rectal palpation; S = Willis dropped ovary technique spay; SM = Willis dropped ovary technique spay with intra-operative delivery of meloxicam; STA = Willis dropped ovary technique spay with intra-operative delivery of topical anaesthetic; and STAM = Willis dropped ovary technique spay with intra-operative delivery of topical anaesthetic and meloxicam. *p*-value for the main effect of treatment. ^A,B,C,D^ Values within a row that do not share a superscript differ significantly at *p* ≤ 0.05. Proportion of time (%) not included for timepoints where no significant effects of treatment on behaviours were observed (*p* > 0.05).

**Table 5 animals-10-00249-t005:** Proportion of time spent exhibiting various back and tail positions or movements (±SE) by heifers in each treatment group at 1–6 h and 24 h post-procedure.

Behaviour	Time (h)	Proportion of Time (%)					*p*-Value
		CON	S	SM	STA	STAM	
Back arch	1–6	2.04 ^A^ ± 0.27	4.46 ^B^ ± 0.33	1.99 ^A^ ± 0.38	5.63 ^B^ ± 0.31	4.57 ^B^ ± 0.33	<0.001
	24						0.761
Relaxed tail	1–6	81.43 ^A^ ± 0.10	67.85 ^BC^ ± 0.13	67.57 ^BC^ ± 0.13	69.17 ^B^ ± 0.13	62.78b ^C^ ± 0.13	<0.001
	24	93.55 ^A^ ± 0.37	75.45 ^BC^ ± 0.43	70.91 ^B^ ± 0.42	77.50 ^BD^ ± 0.43	84.40 ^CD^ ± 0.45	<0.001
Stiff tail	1–6	5.26 ^A^ ± 0.17	17.69 ^B^ ± 0.20	12.23 ^C^ ± 0.21	20.64 ^B^ ± 0.19	19.56 ^B^ ± 0.20	<0.001
	24	2.42 ^A^ ± 0.58	19.09 ^B^ ± 0.63	9.09 ^CD^ ± 0.67	6.67 ^AD^ ± 0.69	3.67 ^AD^ ± 0.77	<0.001
Repetitive tail movement	1-6	13.31 ^AB^ ± 0.11	14.47 ^AD^ ± 0.16	20.20 ^CE^ ± 0.15	10.19 ^B^ ± 0.17	17.67 ^DE^ ± 0.15	<0.001
	24	4.03 ^A^ ± 0.46	5.45 ^AB^ ± 0.62	20.00 ^C^ ± 0.52	15.83 ^C^ ± 0.52	11.93 ^BC^ ± 0.54	<0.001

CON = rectal palpation; S = Willis dropped ovary technique spay; SM = Willis dropped ovary technique spay with intra-operative delivery of meloxicam; STA = Willis dropped ovary technique spay with intra-operative delivery of topical anaesthetic; and STAM = Willis dropped ovary technique spay with intra-operative delivery of topical anaesthetic and meloxicam. *p*-value for the main effect of treatment. ^A,B,C,D,E^ Values within a row that do not share a superscript differ significantly at *p* ≤ 0.05. Proportion of time (%) not included for timepoints where no significant effects of treatment on behaviours were observed (*p* > 0.05).

**Table 6 animals-10-00249-t006:** Proportion of time spent eating and ruminating (±SE) by heifers in each treatment group at 1–6 h and 24 h post-procedure.

Behaviour	Time (h)	Proportion of Time (%)					*p*-Value
		CON	S	SM	STA	STAM	
Eating	1–6	38.60 ^A^ ± 0.08	19.85 ^B^ ± 0.13	34.14 ^A^ ± 0.11	8.31 ^C^ ± 0.15	24.92 ^D^ ± 0.12	<0.001
	24	44.35 ^A^ ± 0.18	25.46 ^B^ ± 0.28	10.91 ^C^ ± 0.35	23.33 ^BD^ ± 0.28	14.68 ^CD^ ± 0.32	<0.001
Ruminating	1–6	9.94 ^A^ ± 0.13	3.38 ^B^ ± 0.25	5.69 ^C^ ± 0.21	0.14 ^D^ ± 0.94	8.52 ^A^ ± 0.19	<0.001
	24	19.35 ^A^ ± 0.23	2.73 ^B^ ± 0.63	56.36 ^C^ ± 0.30	28.33 ^A^ ± 0.30	43.12 ^C^ ± 0.30	<0.001

CON = rectal palpation; S = Willis dropped ovary technique spay; SM = Willis dropped ovary technique spay with intra-operative delivery of meloxicam; STA = Willis dropped ovary technique spay with intra-operative delivery of topical anaesthetic; and STAM = Willis dropped ovary technique spay with intra-operative delivery of topical anaesthetic and meloxicam. *p*-value for the main effect of treatment. ^A,B,C,D^ Values within a row that do not share a superscript differ significantly at *p* ≤ 0.05.

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
