# Peer review of "Preliminary Investigation to Address Pain and Haemorrhage Following the Spaying of Female Cattle"

_animals, 2020, doi:10.3390/ani10020249_

Round 1
Reviewer 1 Report
I thank the authors for a well-written and stimulating study, and I commend this paper for publication with minimal alteration. The paper also signals an important change in the consideration of pain resulting from such procedures in livestock, a topic comparatively understated in associated works (e.g., Witte, K. de, Jubb, T. & Letchford, P. The Dropped Ovary Technique for Spaying Cattle: A Training Manual (Australian Cattle Veterinarians, 2006)). By the way, this reference should be corrected in the bibliography as T. Jubb and P. Letchford have been omitted [18].
Highlights, for me, were the use of generalized linear modelling (assuming the regression model was logit-linked for a binomial error structure, but this was not stated), the prominent use of behavioural measures, and the comprehensive discussion of factors that did not go to plan (i.e., Heifer 47 outcomes, lowered-head scores in controls), or potential side-effects of centrimide applied internally (line 446).
I have only a few comments:
line 184: ovary "removal". A better term might be "excision", as the ovary is not actually removed from the body but dropped into the abdominal cavity (e.g., line 82).
line 226: "...cattle behaviour expert [2], with definitions..." This should be changed to something like "(P. Letchford, co-author)" as it confuses with the convention of in-text citations such as in the sentence close "...associated with interpretation [20]."
General comments:
It is for the following comments that I scored this paper as 'average' for data presentation.
I think that the data is well-presented in tables, however I think that it would be most useful to the general readership of Animals and other casual readers scanning the paper - to provide a reminder of the definitions. Thus a brief explanation of the terms WDOT (Willis dropped-ovary technique) and TA (topical anaesthetic) in the table titles such as for line 232, or as footnotes for Tables 1, Tables 3-6 for defining CON, S, SM, STA and STAM.
It is also my feeling that the general readership will have a greater "take home message" if the Table data were also provided as graphs with means (+/- SE).
Reviewer 2 Report
Thank you for submitting this well presented and interesting manuscript. I have a few comments:
please include the sample size in the abstract LIne 76 - please alter the definition of multimodal analgesia to say it is the use of analgesic drugs with different mechanisms of action, as opposed to using a local anaesthetic and an NSAID Line 92 - be consistent with the way you describe trisolfen. Sometimes it is a multi functional topical anaesthetic.....(lines 81) and later it is the topical anaesthetic and haemostatic wound dressing (not referring to antiseptic, line 92) and then on line 98 it's just a topical anaesthetic. Given the components of trisolfen and your comment about it reducing haemorrhage (line 98) it'd be best to refer to it the same way each time or describe which component may be responsible for certain side effects (e.g. the anaesthetic component isn't likely to reduce haemorrhage you say on line 98). it'd be worth discussing the implications of the timing of administration of meloxicam. You gave it immediately after the procedure but administering it pre-emptively may be better (as described inMitigation of electroencephalographic and cardiovascular responses to castration in Bos indicus bulls following the administration of either lidocaine or meloxicam.
Lehmann HS, Musk GC, Laurence M, Hyndman TH, Tuke J, Collins T, Gleerup KB, Johnson CB.
Vet Anaesth Analg. 2017 Nov;44(6):1341-1352. doi: 10.1016/j.vaa.2017.04.009. Epub 2017 Aug 24.
PMID: 29169838Similar articles
Select item 289568432.Objective Measures for the Assessment of Post-Operative Pain in Bos indicus Bull Calves Following Castration.
Musk GC, Jacobsen S, Hyndman TH, Lehmann HS, Tuke SJ, Collins T, Gleerup KB, Johnson CB, Laurence M.
Animals (Basel). 2017 Sep 28;7(10). pii: E76. doi: 10.3390/ani7100076.
PMID: 28956843 Free PMC ArticleSimilar articles
Line 315 - the table refers to proportion of time spent doing certain things but your text refers to number of animals doing certain things. please correct.
At the beginning of the discussion it would be worth clarifying which behaviours you are interpreting as being associated with pain.
